# Thirty-day readmission after medical-surgical hospitalization for people who experience imprisonment in Ontario, Canada: A retrospective cohort study

**Fiona G. Kouyoumdjian**[1,2,3]*, **Ji Yun Lee**[4], **Aaron M. Orkin**[5,6,7], **Stephanie Y. Cheng**[2], **Kinwah Fung**[2], **Tim O'Shea**[8], **Gordon Guyatt**[8]

**1** Department of Family Medicine, McMaster University, Hamilton, Canada, **2** ICES, Toronto, Canada, **3** Centre for Urban Health Solutions, St. Michael's Hospital, Toronto, Canada, **4** Faculty of Health Sciences, McMaster University, Hamilton, Canada, **5** Department of Emergency Medicine, St Joseph's Health Centre, Unity Health, Toronto, Canada, **6** Department of Family and Community Medicine, University of Toronto, Toronto, Canada, **7** Dalla Lana School of Public Health, University of Toronto, Toronto, Canada, **8** Department of Medicine, Faculty of Medicine, McMaster University, Hamilton, Canada

* kouyouf@mcmaster.ca

**Data Availability Statement:** We are not able to share data because of restrictions specified in our Research Agreement with the Ministry of

## Abstract

We aimed to compare 30-day readmission after medical-surgical hospitalization for people who experience imprisonment and matched people in the general population in Ontario, Canada. We used linked population-based correctional and health administrative data. Of people released from Ontario prisons in 2010, we identified those with at least one medical or surgical hospitalization between 2005 and 2015 while they were in prison or within 6 months after release. For those with multiple eligible hospitalizations, we randomly selected one hospitalization. We stratified people by whether they were in prison or recently released from prison at the time of hospital discharge. We matched each person with a person in the general population based on age, sex, hospitalization case mix group, and hospital discharge year. Our primary outcome was 30-day hospital readmission. We included 262 hospitalizations for people in prison and 1,268 hospitalizations for people recently released from prison. Readmission rates were 7.7% (95%CI 4.4–10.9) for people in prison and 6.9% (95%CI 5.5–8.3) for people recently released from prison. Compared with matched people in the general population, the unadjusted HR was 0.72 (95%CI 0.41–1.27) for people in prison and 0.78 (95%CI 0.60–1.02) for people recently released from prison. Adjusted for baseline morbidity and social status, hospitalization characteristics, and post-discharge health care use, the HR for 30-day readmission was 0.74 (95%CI 0.40–1.37) for people in prison and 0.48 (95%CI 0.36–0.63) for people recently released from prison. In conclusion, people recently released from prison had relatively low rates of readmission. Research is needed to elucidate reasons for lower readmission to ensure care quality and access.

Community Safety and Correctional Services and in the data sharing agreements of ICES. People who would like access to data from the Ministry should direct requests to Michael Kirk at Michael.kirk@ontario.ca. Access to data at ICES can be granted to those who meet pre-specified criteria for confidential access, available at www.ices.on.ca/. Requests to access ICES data for research purposes may be submitted to ICES' Data and Analytic Services, with information available at http://www.ices.on.ca/DAS and contact: das@ices.on.ca.

**Funding:** FK received funding from Physicians' Services Incorporated Foundation (15-22).

**Competing interests:** The authors have declared that no competing interests exist.

## Introduction

In recent years, there has been a substantial focus on readmission after hospitalization as an indicator of quality of care [1]. Readmission rates may reflect baseline morbidity, the development of new conditions or progression of chronic conditions, access to hospital care, inpatient mortality rates, and post-discharge mortality rates [2–5]. More commonly, however, an elevated rate of readmission is assessed as an indicator of deficiencies in quality of hospital care such as incomplete treatment, medical error, and inadequate discharge planning [2, 6, 7].

People who experience imprisonment face a greater burden of illness on average compared to the rest of the general population, with a higher prevalence of infectious diseases, chronic diseases, and mental health and substance use disorders [8]. The period after release from prison is associated with particularly high morbidity and mortality, with several studies showing increased rates of medical-surgical hospitalization compared to the general population [9–18]. Health care access and quality may be suboptimal for this population while in prison and after prison release [13, 19], with substantial impacts on individual and population health [20].

In this context, the rate of medical-surgical hospital readmission among people who experience imprisonment in comparison with the rest of a population is an important indicator of the quality of hospital care. We aimed to compare 30-day readmission rates after medical-surgical hospital admission between people in prison, people recently released from prison, and matched people in the general population in Ontario, Canada.

## Methods

### Study design

Population-based retrospective cohort study.

### Setting

We conducted this study in the province of Ontario, Canada.

Provincial correctional facilities in Canada house people who are admitted to prison without sentencing or who are sentenced to less than two years in prison, as well as people sentenced to two years or longer prior to being transferred to a federal prison and those in temporary detention for other reasons [21]. In Ontario, provincial prisons are publicly funded and administered. We use the term "provincial prison" to represent all provincial correctional facilities, including jails, detention centres, and correctional centres. For Ontario residents, hospitalizations and medically necessary physician services including primary care and emergency department visits are paid for through the public health insurance system, the Ontario Health Insurance Plan, including while in provincial prison.

### Exposure group

We accessed data from the Ontario Ministry of Community Safety and Correctional Services on all adults who were released from provincial prison in 2010. As described elsewhere [13], these data were linked using unique encoded identifiers with health administrative data at ICES, an independent, non-profit organization funded by the Ontario Ministry of Health and Long-Term Care, which houses health administrative data for Ontario residents.

We included all people in the prison group who were hospitalized for a medical or surgical reason during their time in provincial prison or in the 6 months after prison release, were alive on hospital discharge, and were Ontario Health Insurance Plan-eligible for 30 days post-discharge. We selected a 6-month follow-up period after release given our focus on opportunities to improve care in prison and at the time of release and evidence regarding discrimination at

the time of prison release [19, 22]. As health status and health care quality and access may vary substantially in prison and after release, we stratified people who had experienced imprisonment by whether they were discharged from hospital to prison or to the community into groups called *people in prison* and *people recently released from prison.*

We identified hospitalizations in the Canadian Institute for Health Information Discharge Abstracts Database. We included only medical-surgical hospitalizations that were not related to pregnancy or psychiatric causes, as we hypothesized that the mechanisms underlying readmission to hospital for pregnancy and psychiatric reasons may differ. For the same reason, we excluded people who were readmitted for psychiatric or pregnancy reasons. For people in the prison groups who had more than one hospitalization during the follow-up period, we randomly selected one hospitalization as the index hospitalization in order to define the average risk of the outcome of readmission for any single hospitalization, rather than selecting the first or last hospitalization during the follow-up period.

We identified matched people in the general population from Ontario Health Insurance Plan-eligible people in the Registered Persons Database, which is a comprehensive registry of all people with current or prior Ontario Health Insurance Plan coverage [23]. We exactly matched each person in each of the two prison groups with one person in the general population based on sex, birth year, the year of discharge, and the case mix group for the index hospitalization, which indicates a person with similar clinical and heath resource use characteristics [24]. Therefore, we defined four exposure groups: people in prison, people in the general population matched to people in prison, people recently released from prison, and people in the general population matched to people recently released from prison.

## Covariates

We selected indicators of patient socioeconomic status and morbidity, and inpatient and outpatient care, given the role of these factors in readmission and based on available data [5]. We examined age (median and interquartile range, IQR) and sex from the Registered Persons Database, and self-reported race from Ministry of Community Safety and Correctional Services data for those in the prison groups only, as data on race were not available for the general population. We accessed neighbourhood-level data on income quintile using the postal code from Registered Persons Database data at the time of prison release or at the time of hospital admission for people in the general population. We used the Johns Hopkins Adjusted Clinical Group System [25] with a two year look back to determine the number of Aggregated Diagnosis Groups (ADGs), which represent 32 diagnosis clusters that indicate the burden of morbidity [26]; we calculated a summary score as the total number of clusters per person. We applied definitions from the Ontario Mental Health and Addictions Scorecard and Evaluation Framework to identify people with a diagnosis (yes/no) of mood disorders, schizophrenia, anxiety disorders, and substance-related disorders, based on billings in the past two years in the Ontario Health Insurance Plan database, Discharge Abstracts Database, or the Canadian Institute for Health Information National Ambulatory Care Reporting System [27].

We accessed data on length of stay in hospital in days and whether the patient left hospital against medical advice during the index hospitalization (yes/no) from the Discharge Abstracts Database. To examine access to care after hospital discharge, we accessed Ontario Health Insurance Plan data on primary care use (yes/no) and National Ambulatory Care Reporting System data on emergency department use (yes/no) in the 7 days after discharge, which could indicate appropriate post-discharge care or seeking care for unmet care needs, and in the 30 days after discharge.

## Outcome

*A priori*, we defined the primary outcome of interest as 30-day medical-surgical readmission to hospital. We selected the follow up period of 30 days given recent scientific and policy focus on this outcome, including as an indicator of quality of hospital care [1, 2, 5]. We identified readmissions in Discharge Abstracts Database data.

## Sample size

In exploratory analyses as part of a larger project on health care utilization of people released from provincial prison in 2010 [13], we identified 466 medical-surgical hospitalizations in women and 1,722 in men from 2005 to 2015 in prison or in the 6 months after release. Due to repeat hospitalizations and hospitalizations for pregnancy-related causes, we expected there would be fewer total people with medical-surgical hospitalizations not related to pregnancy. Based on a study of another marginalized population in Ontario: people who were homeless [28], we expected that readmission rates would be as high as 22% for the prison group and 17% for the general population. Under those assumptions, we would have greater than 80% power to define readmission frequency for the prison group with a precision of 3% and 95% confidence with 733 people, and greater than 80% power to detect a difference in frequency of up to 5% with a two-sided alpha of 0.05 with 985 people in each exposure group.

## Analyses

We compared people in prison and people recently released from prison, respectively, with general population controls across indicators of health status, socioeconomic status, inpatient care, and post-discharge care, using standardized differences, which are less sensitive to sample size than traditional hypothesis tests; we considered a difference of 10% meaningful [29]. We calculated the number and percent of deaths, admissions for pregnancy, and admissions for psychiatric reasons in the 30 days post-release in exposure groups, as these outcomes may compete with medical-surgical readmission.

We used the Kaplan-Meier method to calculate the frequency of readmission at 30 days after hospital discharge for each exposure group. We censored follow up at the earliest of death, readmission for psychiatric reasons, readmission for pregnancy reasons, or 30 days after hospital discharge [30]. We compared the risk of readmission at 30 days for people in prison and people recently released from prison, respectively, with people in the general population using stratified log-rank tests.

We assessed whether the proportional hazards assumption was met, and then used Cox survival analysis to assess the unadjusted association between imprisonment status and readmission. In adjusted models, we controlled for indicators of baseline health and socio-economic status (neighbourhood income quintile, ADGs), hospitalization (leaving against medical advice, length of stay), and post-discharge care (primary care and emergency department visits) to examine the residual effect of imprisonment status on readmission for people in prison and people recently released from prison, respectively, compared to the general population group.

We used SAS Enterprise Guide version 7.1 for matching and for all analyses.

## Ethics review

This study was reviewed and approved by the Hamilton Integrated Research Ethics Board (#4422). We accessed nominal data only for the purposes of data linkage. We did not obtain informed consent from participants, since the study met criteria for a waiver of consent as per

the Canadian Tri-Council Policy Statement on Ethical Conduct for Research Involving Humans [31], which was approved by the Research Ethics Board.

## Results

We identified 1,670 people who had 1 or more medical-surgical hospitalizations while in prison or in the 6 months after prison release. We excluded 11 people who were not eligible for Ontario Health Insurance Plan coverage for 30 days post-discharge, leaving 1,659 people. We identified matches in the general population for 1,548 people: 262 of whom were discharged from hospital to prison, whom we called *people in prison*, and 1,286 of whom were discharged from hospital to the community, whom we called *people recently released from prison*.

People in prison and people recently released from prison were more likely than people in the general population to live in a neighbourhood in the lowest income quintile, and had greater morbidity at the time of the index hospitalization, as indicated by the median number of ADGs and certain mental disorder diagnoses (Table 1). For people recently released from prison, a higher proportion left against medical advice in the index hospitalization compared with matched people in the general population. People in prison had a longer median length of stay in the index hospitalization compared with matched people in the general population. In the 7 and 30 days after hospital discharge and compared with matched people in the general population, people in prison were more likely to access primary care and people recently released from prison were more likely to access emergency department care, respectively.

Common case mix groups for the index hospitalization were related to substance use, injury, seizures, diabetes, and infection (S1 Table).

Compared with matched people in the general population, readmission rates at 30 days were not significantly different for people in prison, but were significantly lower for people recently released from prison (Table 2 and Fig 1).

In both adjusted and unadjusted models, there was no significant difference in the hazard of readmission between people in prison and matched people in the general population (Table 3). Comparing people recently released from prison with matched people in the general population, there was no significant difference in unadjusted analyses, but the hazard of readmission was significantly lower for people recently released from prison after adjusting for baseline health and social status, hospitalization characteristics, and post-discharge medical care.

## Discussion

This study shows that compared to matched people in the general population, people recently released from prison were 22% less likely to be readmitted to hospital—an absolute decrease of 1.9%. This association persisted after controlling for socio-economic status, morbidity, hospitalization characteristics, and follow up care, with an adjusted hazard ratio of 0.48 for readmission. People in prison were 28% less likely to be readmitted to hospital than matched people in the general population- an absolute decrease of 3.1%- however, this difference was not statistically significant, including after adjustment for covariates in multivariable models.

This study has several strengths. We used population-based data for a large sample of people with current or recent imprisonment, and we matched on variables that would be strongly associated with readmission and that we did not want to explore. No prior study has examined readmission to hospital or important characteristics of hospitalization such as leaving hospital against medical advice for this population.

Potential study limitations are that we included only one hospitalization per person during the period of follow up because of methodological challenges in identifying controls that were

**Table 1. Characteristics of study participants at the time of admission for medical-surgical hospitalization between 2005 and 2015 in Ontario, Canada, by imprisonment status on hospital discharge*†.**

| | | | People in prison, N = 262 | General population matched to people in prison, N = 262 | Standardized difference | People recently released from prison, N = 1,286 | General population matched to people recently released from prison, N = 1,286 | Standardized difference |
|---|---|---|---|---|---|---|---|---|
| Socio-demographic status | Age- median (IQR) years | | 40 (29–47) | 39 (28–47) | 0.00 | 39 (28–48) | 40 (28–48) | 0.00 |
| | Sex | Male | 246 (93.9%) | 246 (93.9%) | 0.00 | 1,069 (83.1%) | 1,069 (83.1%) | 0.00 |
| | Self-reported race§ | Aboriginal | 42 (16.0%) | - | - | 185 (14.4%) | - | - |
| | | Black | 28 (10.7%) | - | - | 80 (6.2%) | - | - |
| | | White | 162 (61.8%) | - | - | 832 (64.7%) | - | - |
| | | Other | 17 (6.5%) | - | - | 104 (8.1%) | - | - |
| | | Missing | 13 (5.0%) | - | - | 85 (6.6%) | - | - |
| | Neighbourhood income quintile | 1st (lowest) | 88 (33.6%) | 64 (24.4%) | 0.20 | 518 (40.3%) | 293 (22.8%) | 0.38 |
| | | 2nd | 49 (18.7%) | 52 (19.8%) | 0.03 | 243 (18.9%) | 277 (21.5%) | 0.07 |
| | | 3rd | 64 (24.4%) | 56 (21.4%) | 0.07 | 218 (17.0%) | 251 (19.5%) | 0.07 |
| | | 4th | 29 (11.1%) | 43 (16.4%) | 0.16 | 151 (11.7%) | 247 (19.2%) | 0.21 |
| | | 5th (highest) | 29 (11.1%) | 45 (17.2%) | 0.18 | 129 (10.0%) | 207 (16.1%) | 0.18 |
| Morbidity | ADGs‖ | Median (IQR) | 8 (6–11) | 6 (4–10) | 0.44 | 9 (6–12) | 7 (5–10) | 0.36 |
| | Mental illness | Mood disorders | 27 (10.3%) | 14 (5.3%) | 0.19 | 225 (17.5%) | 109 (8.5%) | 0.27 |
| | | Schizophrenia | 12 (4.6%) | ‡ | 0.15 | 95 (7.4%) | 32 (2.5%) | 0.23 |
| | | Anxiety disorders | 25 (9.5%) | 17 (6.5%) | 0.11 | 220 (17.1%) | 93 (7.2%) | 0.31 |
| | | Substance-related disorders | 85 (32.4%) | 49 (18.7%) | 0.32 | 556 (43.2%) | 220 (17.1%) | 0.59 |
| Index hospital admission | Left against medical advice | | 7 (2.7%) | 9 (3.4%) | 0.04 | 122 (9.5%) | 49 (3.8%) | 0.23 |
| | Length of stay | Median (IQR) days | 3 (1–6) | 3 (1–5) | 0.19 | 3 (1–6) | 3 (1–5) | 0.07 |
| | | <2 days | 68 (26.0%) | 85 (32.4%) | 0.14 | 391 (30.4%) | 408 (31.7%) | 0.03 |
| | | 2–4 days | 97 (37.0%) | 100 (38.2%) | 0.02 | 471 (36.6%) | 491 (38.2%) | 0.03 |
| | | 5–9 days | 64 (24.4%) | 54 (20.6%) | 0.09 | 231 (18.0%) | 250 (19.4%) | 0.04 |
| | | ≥10 days | 33 (12.6%) | 23 (8.8%) | 0.12 | 193 (15.0%) | 137 (10.7%) | 0.13 |
| Post-discharge care | Primary care | 7 days | 181 (69.1%) | 64 (24.4%) | 1.00 | 329 (25.6%) | 348 (27.1%) | 0.03 |
| | | 30 days | 224 (85.5%) | 125 (47.7%) | 0.87 | 637 (49.5%) | 677 (52.6%) | 0.06 |
| | Emergency department care | 7 days | 37 (13.1%) | 29 (11.1%) | 0.09 | 223 (17.3%) | 151 (11.7%) | 0.16 |
| | | 30 days | 71 (27.1%) | 68 (26.0%) | 0.03 | 444 (34.5%) | 304 (23.6%) | 0.24 |
| Competing outcomes | Death | 30 days | ‡ | ‡ | 0.09 | 10 (0.8%) | 12 (0.9%) | 0.02 |
| | Psychiatric admission | 30 days | ‡ | ‡ | 0.12 | 29 (2.3%) | 23 (1.8%) | 0.03 |
| | Pregnancy admission | 30 days | 0 (0%) | 0 (0%) | 0.00 | 0 (0%) | 0 (0%) | 0.00 |

*People in prison were discharged from hospital while in provincial prison. People recently released from prison were discharged from hospital to the community within 6 months of release from provincial prison. The general population group was people who were matched by age, sex, case mix group, and discharge year to people in prison and people recently released from prison.

†n (%) unless otherwise indicated.

‡For cells with n ≤5, we suppressed the number as per ICES policy. Schizophrenia and psychiatric readmission were each significantly more common in people in prison compared with matched people in the general population.

§Data on race were not available for general population controls.

‖ADGs = Aggregated Diagnosis Groups from the Johns Hopkins Adjusted Clinical Group System.

**Table 2. Readmission by 30 days after medical-surgical hospitalization for people in prison and people recently released from prison,\* and age-, sex-, and case mix group-matched people in the general population, from Kaplan-Meier analyses†.**

| Exposure group | % (95% CI) | p value‡ |
|---|---|---|
| People in prison, N = 262 | 7.7 (4.4, 10.9) | 0.46 |
| General population matched to people in prison, N = 262 | 10.8 (7.0, 14.5) | |
| People recently released from prison, N = 1,286 | 6.9 (5.5, 8.3) | 0.04 |
| General population matched to people recently released from prison, N = 1,286 | 8.8 (7.2, 10.3) | |

\*People released from provincial prison in Ontario in 2010 who were admitted to hospital between 2005 and 2015 while in provincial prison or within 6 months of release from provincial prison.

†Follow up period was censored for death, or hospital admission for psychiatric or pregnancy reasons.

‡From stratified log-rank test.

matched on the basis of case mix group for each hospitalization. Depending on the patient's prior experience, any single hospitalization may in fact represent a readmission as opposed to an initial hospitalization. Since hospitalization occurs more frequently in people who experience imprisonment [13], readmissions may be overrepresented in people in prison and people recently released from prison compared to the general population group. The random selection of a single hospitalization allowed us, however, to define the average risk of the outcome of readmission for each hospitalization. As people who were in prison at the time of hospital discharge may have been released from prison over the 30 day follow up period and people who were recently released at the time of discharge may have been readmitted to prison during the 30 day follow up period, some people may have spent time in prison and in the community

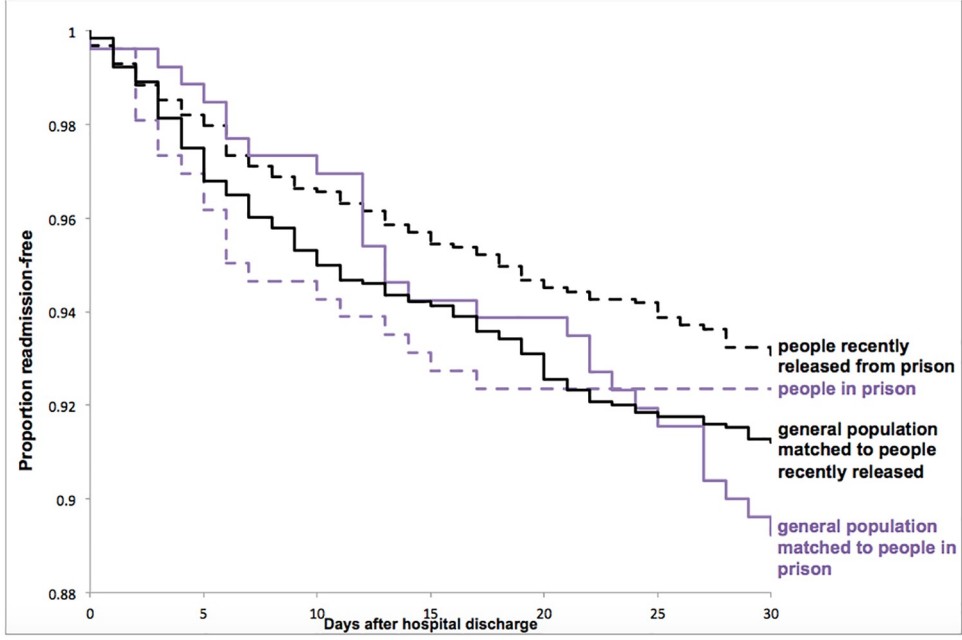

**Fig 1. Kaplan-Meier curves\* for readmission by 30 days after medical-surgical hospitalization for people in prison, people recently released from prison,† and age-, sex-, and case mix group-matched people in the general population in Ontario, Canada.** \*Follow up was censored for death, or hospital admission for psychiatric or pregnancy reasons. †People released from provincial prison in Ontario in 2010 who were admitted to hospital between 2005 and 2015 while in provincial prison or within 6 months of release from provincial prison.

**Table 3. Hazard ratios from Cox proportional hazards models for readmission by 30 days after medical-surgical hospitalization for people in prison and people recently released from prison* compared with age-, sex-, case mix group-matched people in the general population in Ontario, Canada.**

| Variables adjusted for in model | | People in prison, N = 262 | People recently released from prison, N = 1,286 |
|---|---|---|---|
| None | | 0.72 (0.41–1.27) | 0.78 (0.60–1.02) |
| Socio-economic status and morbidity | A) Neighbourhood income quintile | 0.72 (0.40–1.29) | 0.78 (0.60–1.03) |
| | B) ADGs† | 0.60 (0.34–1.07) | 0.65 (0.49–0.85) |
| | A and B | 0.62 (0.35–1.12) | 0.65 (0.59–0.86) |
| Index hospital admission | C) Left against medical advice | 0.73 (0.42–1.28) | 0.73 (0.56–0.95) |
| | D) Length of stay | 0.69 (0.39–1.20) | 0.76 (0.58–0.99) |
| | C and D | 0.68 (0.39–1.19) | 0.71 (0.54–0.93) |
| Medical care post-hospital discharge | E) Primary care | 0.74 (0.40–1.38) | 0.78 (0.60–1.02) |
| | F) Emergency department care | 0.71 (0.40–1.26) | 0.54 (0.41–0.70) |
| | E and F | 0.69 (0.38–1.25) | 0.54 (0.42–0.71) |
| All (A, B, C, D, E, and F) | | 0.74 (0.40–1.37) | 0.48 (0.36–0.63) |

*People released from provincial prison in Ontario in 2010 who were admitted to hospital between 2005 and 2015 while in provincial prison or within 6 months of release from provincial prison.

†ADGs = Aggregated Diagnosis Groups from the Johns Hopkins Adjusted Clinical Group System.

over the 30 day follow up period after hospital discharge. This may have contributed to exposure misclassification bias. Since we matched on case mix group, the general population groups were likely enriched for people who were more similar in terms of medical illnesses and risk behaviours compared to a general population group matched only on socio-demographic factors, for example, people with substance use disorders. In this way, to the extent that these factors impact readmission, we would expect less difference in outcomes between groups than if we had not matched based on case mix group. As there were only 262 people in the group of people in prison, analyses involving this group may not have had adequate power, so these results should be considered exploratory. Finally, given our use of administrative data rather than data collected data from patients, providers or charts, we were not able to define the reasons for readmission, *i.e.* to what extent readmission reflects inadequate care during the initial hospitalization, initial follow up on discharge from hospital, or patient-specific factors such as morbidity or behaviours.

The findings of our study largely agree with previous evidence regarding increased morbidity in this population [8, 32]. While studies on other marginalized populations with increased morbidity and barriers to care access such as people who are homeless and people with developmental disabilities have identified higher rates of readmission to hospital [28, 33], we found no difference between rates of readmission for people in prison and we found decreased rates of readmission for people recently released from prison compared to the general population.

While the absolute difference in readmission rates between people recently released from prison and matched people in the general population was small, at 1.9%, we found the lower rate surprising; we had expected the readmission rate to be higher for people recently released from prison given evidence regarding their relatively high morbidity [8, 32]. There are several potential explanations for the lower readmission rate in people recently released from prison compared with matched people in the general population. This population may be receiving better care in hospital and after hospital discharge. We found a similar length of stay in hospital (median of 3 days (IQR 1–6) vs. 3 days (IQR 1–5)) and a higher proportion of people who left against medical advice (9.5% vs. 3.8%); these data do not support the hypothesis of better inpatient care. People recently released from prison did not have higher rates of follow up with primary care in the week after hospital discharge (25.6% vs. 27.1%), and in fact, primary care use

in people recently released from prison may be more likely to reflect use of specific services such as addictions clinics rather than generalist care, given the relatively high proportion of people in this group with substance-related disorders. Regarding morbidity and socio-economic status, those recently released from prison had a higher number of ADGs and a higher proportion of people in the lowest neighbourhood income quintile, and these factors are usually associated with increased readmission risk [5]. Competing outcomes such as death or hospital admission for psychiatric or pregnancy reasons were relatively uncommon, and would not explain the difference between groups. It is possible that people recently released from prison were able to get their needs met through primary care and ED encounters without needing admission, and ED visits after hospital discharge were more common in people who were recently released compared with matched people in the general population. Even after controlling for all these factors in multivariable models, however, the readmission risk remained significantly lower for people recently released from prison, though this may be due to not having adequately controlled for these variables or to additional factors related to imprisonment status.

Given increased emergency department visits, it is also possible that people recently released from prison had clinical indications for readmission, but either were not offered readmission or were offered but did not accept the offer of readmission. This is concerning, as it could indicate discrimination on the basis of legal status [19, 34–36] or on the basis of characteristics that are overrepresented in this population such as low socioeconomic status or mental illness. If people were choosing to not follow medical advice when they are acutely ill, this would also be problematic; deciding to not be admitted to hospital could reflect competing priorities such as the need to use substances, considering the high prevalence of substance use disorders and admissions for substance-use related conditions [14–18, 37, 38].

For people in prison, factors affecting readmission differ compared with both the general population and people recently released from prison. We found that people in prison had similar multimorbidity but a lower prevalence of mental illness compared with people recently released from prison, and greater multimorbidity and mental illness prevalence compared with the general population. Length of the index hospitalization was similar across groups, and the proportion that left against medical advice was similar between people in prison and the general population. In prison, policies and procedures affect health care access, for example the high proportion of people who accessed primary care (69.1% at 7 days post-discharge compared to 24.4% for people in the general population) likely reflects the policy of routine physician follow up in prison after hospital discharge, and health care providers and correctional officers in prison act as gatekeepers to people leaving prison to access the emergency department. Also, correctional officers accompany people in prison when they access care in the emergency department, which may affect health care provider and patient decisions regarding admission.

Further research is required to elucidate the experiences of people in prison and after prison release with hospitalization, and in particular to understand why rates of readmission are lower for people recently released from prison. Attention should be paid to barriers to hospital access on the provider or patient side, as well as to defining ways to optimize primary care and emergency department care to better meets the needs of this population. The events around visits to the emergency department, and the decision to admit or not to admit, might provide particular insights.

## Supporting information

**S1 Table. Most common case mix groups for the index hospitalization among prison group,** * **N = 1,548.**
(DOCX)

**S2 Table. STROBE statement.**
(DOC)

## Acknowledgments

Disclaimer: Parts of this material are based on data and/or information compiled and provided by the Canadian Institute for Health Information. However, the analyses, conclusions, opinions and statements expressed in the material are those of the authors, and not necessarily those of the Canadian Institute for Health Information.

This study was supported by ICES, which is funded by an annual grant from the Ontario Ministry of Health and Long-Term Care. The opinions, results and conclusions reported in this paper are those of the authors and are independent from the funding sources. We acknowledge the Ontario Ministry of Community Safety and Correctional Services, which provided data for the study. No endorsement by ICES, the Ontario Ministry of Health and Long-Term Care, or the Ministry of Community Safety and Correctional Services is intended or should be inferred.

## Author Contributions

**Conceptualization:** Fiona G. Kouyoumdjian, Ji Yun Lee, Aaron M. Orkin, Tim O'Shea, Gordon Guyatt.

**Formal analysis:** Stephanie Y. Cheng, Kinwah Fung.

**Funding acquisition:** Fiona G. Kouyoumdjian.

**Methodology:** Ji Yun Lee, Aaron M. Orkin, Stephanie Y. Cheng, Tim O'Shea, Gordon Guyatt.

**Supervision:** Fiona G. Kouyoumdjian.

**Writing – original draft:** Fiona G. Kouyoumdjian, Ji Yun Lee.

**Writing – review & editing:** Aaron M. Orkin, Stephanie Y. Cheng, Kinwah Fung, Tim O'Shea, Gordon Guyatt.

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
