## [Decision Letter · Decision Letter 0]

12 Sep 2019

PONE-D-19-17979

Thirty-day readmission and return to acute care after hospitalization for people who experience imprisonment in Ontario, Canada: A retrospective cohort study

PLOS ONE

Dear Dr. Kouyoumdjian,

Thank you for submitting your manuscript to PLOS ONE. After careful consideration, we feel that it has merit but does not fully meet PLOS ONE’s publication criteria as it currently stands. Therefore, we invite you to submit a revised version of the manuscript that addresses the points raised during the review process.

We would appreciate receiving your revised manuscript by Oct 27 2019 11:59PM. To enhance the reproducibility of your results, we recommend that if applicable you deposit your laboratory protocols in protocols.io, where a protocol can be assigned its own identifier (DOI) such that it can be cited independently in the future. For instructions see: http://journals.plos.org/plosone/s/submission-guidelines#loc-laboratory-protocols

We look forward to receiving your revised manuscript.

Kind regards,

Lars-Peter Kamolz, M.D., Ph.D., M.Sc.

Academic Editor

PLOS ONE

Journal Requirements:

2. In ethics statement in the manuscript and in the online submission form, please provide additional information about the patient records used in your retrospective study. Specifically, please ensure that you have discussed whether all data were fully anonymized before you accessed them and/or whether the IRB or ethics committee waived the requirement for informed consent. If patients provided informed written consent to have data from their medical records used in research, please include this information.

3. Please correct your reference to "p=0.000" to "p<0.001" or as similarly appropriate, as p values cannot equal zero.

4. We noticed you have some minor occurrence of overlapping text with the following previous publication(s), which needs to be addressed:

- Kouyoumdjian, Fiona G., et al. "The health care utilization of people in prison and after prison release: A population-based cohort study in Ontario, Canada." PloS one 13.8 (2018): e0201592.

Khanna, Sumeet, et al. "Health care utilization by people with HIV on release from provincial prison in Ontario, Canada in 2010: a retrospective cohort study." AIDS care 31.7 (2019): 785-792.

 The text that needs to be addressed involves the first paragraph of the Introduction.

In your revision ensure you cite all your sources (including your own works), and quote or rephrase any duplicated text outside the methods section. Further consideration is dependent on these concerns being addressed.

"This study was supported by ICES, which is funded by an annual grant from the Ontario Ministry of Health and Long-Term Care. The opinions, results and conclusions reported in this paper are those of the authors and are independent from the funding sources. We  acknowledge the Ontario Ministry of Community Safety and Correctional Services, which provided data for the study. No endorsement by ICES, the Ontario Ministry of Health and Long-Term Care, or the Ministry of Community Safety and Correctional Services is intended or should

be inferred. Parts of this material are based on data and/or information compiled and provided by

CIHI. However, the analyses, conclusions, opinions and statements expressed in the material are

those of the authors, and not necessarily those of CIHI.

"FK received funding from Physicians' Services Incorporated Foundation (15-22)."

Reviewers' comments:

Reviewer's Responses to Questions

**Comments to the Author**

1. Is the manuscript technically sound, and do the data support the conclusions?

Reviewer #1: Partly

Reviewer #2: Yes

2. Has the statistical analysis been performed appropriately and rigorously? 

Reviewer #1: Yes

Reviewer #2: Yes

3. Have the authors made all data underlying the findings in their manuscript fully available?

Reviewer #1: No

Reviewer #2: No

4. Is the manuscript presented in an intelligible fashion and written in standard English?

Reviewer #1: Yes

Reviewer #2: Yes

5. Review Comments to the Author

Reviewer #1: This manuscript investigates the readmission and return to acute care after hospitalization among persons who experienced incarcerations and persons from the general population. This is an important research question, as the released prisoners constitute an underserved population and because their health care use is not well understood. The manuscript relies on a large sample size with a sound methodology. However, there are some limitations that should be addressed.

Whole paper

1. The literature more commonly uses “emergency department visits” rather than “return to acute care”.

Abstract

2. I found the abstract very confusing at first read. I am afraid it is not readable as a stand-alone piece. It should be written more clearly. The results section is very long and the conclusions are almost absent. I did not understand the Methods at first read.

Introduction

3. The introduction is very short. There are some studies addressing health care (and use of emergency department) among (released) prisoners. This literature should be presented as well.

As it is not possible to understand the reasons of readmission or return to acute care, I suggest to avoid mentioning it at the first place in the Introduction.

Methods

4. The most important shortcoming is the presence of the incarcerated prisoners and the potential lack of power. The non-significant findings are not interpretable and there are important issues (prisoners possibly released). I suggest to remove this group from the sample and to focus on released prisoners and general population.

5. What about released prisoners who are reincarcerated within 6 months?

6. Using a random hospitalization appears as a limitation of the study design. The rationale for this choice instead of using the first hospitalization should be given.

7. Please add detailed information on variables assessed in the study.

8. Please also give a rationale for using a 30-day readmission or return to acute care cut-off.

Results

9. The 7-day outcome is mentioned for the first time p. 9 in the Results section. It should be included in the study’s objective with its rationale and in the Methods section.

10. An important missed variable is the reason for hospitalization. It should be controlled for to achieve a better understanding of the results.

Discussion

11. Released prisoners are more likely to visit emergency departments in comparison with the general population. This may be because they have no GP, no information on their own health insurance, or no money to pay extra fees. It does not mean that they have worse outcomes after hospitalization. This should be discussed in the paper. Information on health care fees in the Canadian health care system should also be included.

Reviewer #2: This study is interesting for several reasons:

- large samples of people

- lack of data on care and hospitalizations for people in prison and after release

- comparison with the general population

However, for the method, why the 3 groups do not have the same number of people, since the objective was to compare them.

For the results, it is a little difficult to understand why there is such a difference between readmission and return to acute care. Information on the reasons for using non-hospitalized emergencies would deserve to be developed. Interpretation is possibly very much related to the functioning of the health care system in Ontario.

Minor comments:

Introduction

- Line 62: abreviation “emergency department (ED)” already indicated line 46

Methods

- Line 84: define ICES

6. PLOS authors have the option to publish the peer review history of their article (what does this mean?). If published, this will include your full peer review and any attached files.

Reviewer #1: No

Reviewer #2: No

---

## [Author Response · Author response to Decision Letter 0]

22 Oct 2019

Please see the detailed responses provided in the response to reviewers document.

---

## [Decision Letter · Decision Letter 1]

19 Nov 2019

PONE-D-19-17979R1

Thirty-day readmission after medical-surgical hospitalization for people who experience imprisonment in Ontario, Canada: A retrospective cohort study

PLOS ONE

Dear Dr. Kouyoumdjian,

Thank you for submitting your manuscript to PLOS ONE. After careful consideration, we feel that it has merit but does not fully meet PLOS ONE’s publication criteria as it currently stands. Therefore, we invite you to submit a revised version of the manuscript that addresses the points raised during the review process.

We would appreciate receiving your revised manuscript by Jan 03 2020 11:59PM. To enhance the reproducibility of your results, we recommend that if applicable you deposit your laboratory protocols in protocols.io, where a protocol can be assigned its own identifier (DOI) such that it can be cited independently in the future. For instructions see: http://journals.plos.org/plosone/s/submission-guidelines#loc-laboratory-protocols

We look forward to receiving your revised manuscript.

Kind regards,

Lars-Peter Kamolz, M.D., Ph.D., M.Sc.

Academic Editor

PLOS ONE

Reviewers' comments:

Reviewer's Responses to Questions

**Comments to the Author**

1. If the authors have adequately addressed your comments raised in a previous round of review and you feel that this manuscript is now acceptable for publication, you may indicate that here to bypass the “Comments to the Author” section, enter your conflict of interest statement in the “Confidential to Editor” section, and submit your "Accept" recommendation.

Reviewer #1: (No Response)

Reviewer #2: All comments have been addressed

2. Is the manuscript technically sound, and do the data support the conclusions?

Reviewer #1: Yes

Reviewer #2: Partly

3. Has the statistical analysis been performed appropriately and rigorously? 

Reviewer #1: Yes

Reviewer #2: Yes

4. Have the authors made all data underlying the findings in their manuscript fully available?

Reviewer #1: Yes

Reviewer #2: No

5. Is the manuscript presented in an intelligible fashion and written in standard English?

Reviewer #1: Yes

Reviewer #2: Yes

6. Review Comments to the Author

Reviewer #1: The authors addressed most of my comments and the current version is improved. I still have some comments, listed below (page numbers refer to the manuscript with track change).

p. 8 lines 131-138: Please explain how you matched groups (e.g., software).

Table 1: There is a mistake with the sample size of the matched group for released prisoners (n=1,286 and not n=1,548).

p. 8 line 134: Please define what is case-mix group of the index hospitalization.

Information is still needed on the variables: ADG and Ontario Mental Health and Addictions Scorecard and Evaluation Framework: are those variable summary scores? Continuous? What range? Etc. Please explain more clearly that primary care and ED use are variables (and how they were assessed).

Please avoid using too many abbreviations, the manuscript is quite hard to follow (see for example sentence p.9 line 156-160: 5 abbreviations).

Discussion: Please also discuss the effect size of the decrease in hospital readmission for prisoners recently released from prison. Maybe it is not clinically relevant.

Reviewer #2: You have decided to cut content regarding the 7-day period, since your focus

is on 30-day readmission rates and your regression analyses are focused on the 30-day period, and in

consideration of sample size issues. You must cut the 7-day data in Table 1 in post discharge section (primary care and emergency department care) and line 401.

7. PLOS authors have the option to publish the peer review history of their article (what does this mean?). If published, this will include your full peer review and any attached files.

Reviewer #1: No

Reviewer #2: No

---

## [Author Response · Author response to Decision Letter 1]

27 Nov 2019

Please see attached response to reviewers.

---

## [Decision Letter · Decision Letter 2]

23 Dec 2019

Thirty-day readmission after medical-surgical hospitalization for people who experience imprisonment in Ontario, Canada: A retrospective cohort study

PONE-D-19-17979R2

Dear Dr. Kouyoumdjian,

We are pleased to inform you that your manuscript has been judged scientifically suitable for publication and will be formally accepted for publication once it complies with all outstanding technical requirements.

With kind regards,

Lars-Peter Kamolz, M.D., Ph.D., M.Sc.

Academic Editor

PLOS ONE

Additional Editor Comments (optional):

Reviewers' comments:

Reviewer's Responses to Questions

**Comments to the Author**

1. If the authors have adequately addressed your comments raised in a previous round of review and you feel that this manuscript is now acceptable for publication, you may indicate that here to bypass the “Comments to the Author” section, enter your conflict of interest statement in the “Confidential to Editor” section, and submit your "Accept" recommendation.

Reviewer #1: All comments have been addressed

Reviewer #2: All comments have been addressed

2. Is the manuscript technically sound, and do the data support the conclusions?

Reviewer #1: Yes

Reviewer #2: Yes

3. Has the statistical analysis been performed appropriately and rigorously? 

Reviewer #1: Yes

Reviewer #2: Yes

4. Have the authors made all data underlying the findings in their manuscript fully available?

Reviewer #1: Yes

Reviewer #2: No

5. Is the manuscript presented in an intelligible fashion and written in standard English?

Reviewer #1: Yes

Reviewer #2: Yes

6. Review Comments to the Author

Reviewer #1: (No Response)

Reviewer #2: (No Response)

7. PLOS authors have the option to publish the peer review history of their article (what does this mean?). If published, this will include your full peer review and any attached files.

Reviewer #1: No

Reviewer #2: No

---

## [Editor Report · Acceptance letter]

30 Dec 2019

PONE-D-19-17979R2 

Thirty-day readmission after medical-surgical hospitalization for people who experience imprisonment in Ontario, Canada: A retrospective cohort study 

Dear Dr. Kouyoumdjian:

I am pleased to inform you that your manuscript has been deemed suitable for publication in PLOS ONE. Congratulations! Your manuscript is now with our production department. 

With kind regards,

on behalf of

Dr. Lars-Peter Kamolz 

Academic Editor

PLOS ONE